ecology

stable isotope analysis, sperm whales, Gulf of Alaska, dietary mixing models

**Author for correspondence:**
Lauren A. Wild
e-mail: lawild@alaska.edu

# Exploring variability in the diet of depredating sperm whales in the Gulf of Alaska through stable isotope analysis

Lauren A. Wild[1], Franz Mueter[1], Briana Witteveen[1] and Janice M. Straley[2]

[1]College of Fisheries and Ocean Sciences, University of Alaska Fairbanks, 12101 Point Lena Loop, Juneau, AK 99801, USA
[2]Department of Biology, University of Alaska Southeast, 1332 Seward Ave, Sitka, AK 99835, USA

LAW, 0000-0002-6186-7252

Sperm whales interact with commercially important groundfish fisheries offshore in the Gulf of Alaska (GOA). This study aims to use stable isotope analysis to better understand the trophic variability of sperm whales and their potential prey, and to use dietary mixing models to estimate the importance of prey species to sperm whale diets. We analysed tissue samples from sperm whales and seven potential prey (five groundfish and two squid species). Samples were analysed for stable carbon and nitrogen isotope ratios, and diet composition was estimated using Bayesian isotopic mixing models. Mixing model results suggest that an isotopically combined sablefish/dogfish group, skates and rockfish make up the largest proportion of sperm whale diets (35%, 28% and 12%) in the GOA. The top prey items of whales that interact more frequently with fishing vessels consisted of skates (49%) and the sablefish/dogfish group (24%). This is the first known study to provide an isotopic baseline of adult male sperm whales and these adult groundfish and offshore squid species, and to assign contributions of prey to whale diets in the GOA. This study provides information to commercial fishermen and fisheries managers to better understand trophic connections of important commercial species.

## 1. Introduction

Understanding top predator diets and their role in marine food webs is important to managing fisheries and mammals from an

ecosystem perspective. Sperm whales (*Physeter macrocephalus*) are the largest of the toothed whales, a deep-diving cosmopolitan species inhabiting the world's major oceans. Females and calves primarily inhabit warm equatorial regions between 40°S and 40°N latitude, while males are thought to leave their natal groups after age 12, when they move to high-latitude feeding grounds and roam widely [1,2]. One of those high-latitude foraging grounds is in the Gulf of Alaska (GOA), where sperm whales were historically killed in large numbers during commercial whaling [3,4]. An estimated 157 680 sperm whales were killed in the North Pacific Ocean between 1948 and 1972 by Russian whaling ships alone [4], at a time when there was thought to be a population of 1 260 000 whales in that region [2]. A current reliable population estimate for the entire North Pacific Ocean does not exist, nor does one exist for the GOA itself. Within the GOA, however, a line-transect survey in the western GOA estimated an abundance of 345 (CV = 0.43) sperm whales in 2015 [5], and a mark-recapture abundance estimate for the eastern GOA in 2014 estimated 135 (95% CI: 124–153) individuals [6].

## 1.1. Sperm whale historical diet

Sperm whales primarily consume cephalopods worldwide, but in some parts of the world, including British Columbia, New Zealand and Antarctic waters, fish are a considerable portion of the diet for males [1,7–11]. Stomach content data from scientists on whaling ships in Alaskan waters in the 1960s indicate sperm whale diets were dominated by squid in the Bering Sea and western GOA, but as whaling ships moved into the eastern GOA and off the northern British Columbia coast, stomachs contained a majority of fish remains [8,12–14]. Of these fishes, the most common occurrences in sperm whale stomachs noted in the 'Northeast Pacific' and from northern British Columbia whaling stations between 1936 and 1967 were ragfish (*Icosteus aenigmaticus*), rockfish (*Sebastes* spp.), skates (*Rajidae* spp.) and dogfish (*Squalus suckleyi*) [8,11,14,15]. Of the squid remains collected from sperm whale stomachs in the North Pacific, including the GOA and British Columbia coast, the primary species found were robust clubhook squid (*Onykia robusta*) and magister armhook squid (*Berryteuthis magister*) [11,12,14–16]. Sablefish as a diet item were only mentioned once in stomach contents data from northern California whaling stations [8].

Since the rapid decline in commercial whaling following the International Whaling Commission's moratorium in 1986, studying the diet of large free-ranging cetaceans has become more difficult, as stomach samples from these animals are no longer readily available and are limited to stranded animals, which is rare in a species that generally lives far from shore. In fact, there appear to have been no publications of collections or analysis of stomach contents from stranded sperm whales in the GOA or Northeast Pacific Ocean in the last 40 years. Collection of faeces can give insight into the diet but requires opportunistic detection and collection; a whale has to be observed defecating, and faeces must be collected before it dissipates and sinks. Thus few studies exist for sperm whales that include analysis of faeces, and none that we found for the GOA or Northeast Pacific Ocean. Direct observations of feeding sperm whales are not practical as they forage and consume prey at depth. Overall, while these methods can provide a snapshot of recent diet, they suffer from low sample sizes, high cost of collection, potential bias related to cause of death for stranded animals, and bias towards hard parts from prey species that are less likely to break down in the stomach and can, therefore, be identified in stomachs or faeces.

## 1.2. Sperm whales in the Gulf of Alaska

Today, sperm whales are still found in the GOA, where they are known to remove sablefish (*Anoplopoma fimbria*) from commercial longline fishing vessels [6,17,18]. This removal of caught fish, known as depredation, has been observed worldwide on multiple fish species [19,20], and in the GOA has increased since the implementation of the catch share or individual fishing quota programme in the mid-1990s [6,18]. The economic impacts of depredation to the longline fishing fleet in the GOA are nuanced, and result in added costs for fishermen in fuel, bait, crew and gear to make up for fish lost to whales [6,21,22].

Since 2003 the Southeast Alaska Sperm Whale Avoidance Project (SEASWAP) has been studying sperm whale depredation of longline fishing gear as a collaborative effort between scientists, fishermen and fisheries managers, with the goal to minimize interactions between whales and fishing gear. SEASWAP effort is primarily based out of Sitka, on the outer coast of Southeast Alaska, and most research focuses on the eastern GOA study area, though depredation by sperm whales extends into the central and western GOA as well. Initial work found that whales cue in to the unique patterns of propeller cavitation made by longline vessels as they shift in and out of gear to stay on

top of the line as it is hauled to the surface [23]. Whales can pick up these vessel-hauling sounds acoustically from at least 10–13 km away [24], with anecdotal evidence from research cruises showing detection of vessels on a calm day upwards of 16 km (J Straley 2009, unpublished data). Through acoustic and visual observations, SEASWAP has found that whales surfacing within 500 m of a fishing vessel hauling gear can be considered engaging in depredation activity [25], a metric that has been independently estimated in other parts of the world [19,20,26]. SEASWAP has only observed male sperm whales interacting with longline fishing gear, which has been corroborated by genetic analysis of tissue samples collected from individual whales [27]. This finding is consistent with historical records in which Russian scientists on whaling ships noted male-only bachelor groups in the eastern GOA, along the continental slope [3].

## 1.3. Stable isotopes

Stable isotope analysis has become a useful tool to estimate recent diet composition, trophic position and food web connections, as well as to construct time series of dietary estimates [28–30]. The primary isotopes used in food web studies are carbon and nitrogen. Stable carbon isotope ratios (ratio of $^{13}C$ : $^{12}C$ isotopes in a tissue, with respect to an international standard) reflect photosynthetic pathways of an animal's food sources [29]. Therefore, in the marine environment, stable carbon isotope ratios ($\delta^{13}C$), can show differences between benthic and pelagic foragers, between near shore and offshore foragers and between freshwater and saltwater foragers [30–32]. Additionally, they can reflect latitudinal gradients across ocean basins [33,34]. Stable nitrogen isotope ratios (ratio of $^{15}N$ : $^{14}N$ isotopes in a tissue, with respect to an international standard) are considered a proxy for trophic level, as nitrogen isotopes experience metabolic fractionation as they move up the food chain from prey to predator [29,35,36]. Consequently, a predator's stable nitrogen isotope ratio ($\delta^{15}N$) will be higher than that of its prey in a process known as trophic enrichment.

Tissue-specific stable isotope ratios from predators (mixtures) and their prey (sources) can be used in isotopic mixing models to evaluate the proportional contribution of prey to predator diets [37–41]. For cetacean species, a variety of tissues such as teeth, baleen, skin, muscle and blubber have been sampled and analysed using stable isotope analysis. While baleen, teeth and muscle must be collected post-mortem, and therefore pose the same problems as collection of stomach contents, skin and blubber can be collected from free-ranging animals. Biopsy sampling of cetaceans is a minimally invasive technique that does not require handling or capturing of animals, and results in the collection of a small amount of skin and blubber that can be used for a variety of genetic, metabolomic and isotopic analyses [42,43].

## 1.4. Objectives of present study

The current diet of sperm whales in the GOA is poorly understood. The goal of this project was to better understand the trophic position and foraging ecology of sperm whales and assess the importance of sablefish in modern sperm whale diets. To do this, our main objectives were to: (1) use stable isotope analysis to describe isotopic variability and calculate trophic position of male sperm whales and their prey in the GOA; and (2) use isotopic mixing models to assess the proportion of various prey to sperm whale diets in the GOA, and compare to historical stomach contents data where possible.

# 2. Material and methods

## 2.1. Data collection—whale biopsies

A total of 33 biopsy tissue samples were used in this experiment, collected from sperm whales during SEASWAP research experiments from 2003 to 2017 (figure 1). Three samples, collected from the NOAA GOA longline survey in 2006, were taken in the central GOA, outside the study area (figure 1). However, these samples fell within the range of $\delta^{13}C$ and $\delta^{15}N$ values of samples collected in the eastern GOA and isotope ratios were not significantly different than the rest of samples (MANOVA, $F_{8,56} = 2.19$, $p = 0.05$), and so they were included in the analysis to help improve our sample size. Samples were collected under NOAA research permits nos. 14122 and 18529, and Institutional Animal Care and Use Committee project no. 906340-4. These permits ensure the safe and humane interactions between researchers and animals. Sperm whales were not captured or handled at all during the collection of biopsy tissues, and no animals were harmed or killed during the study.

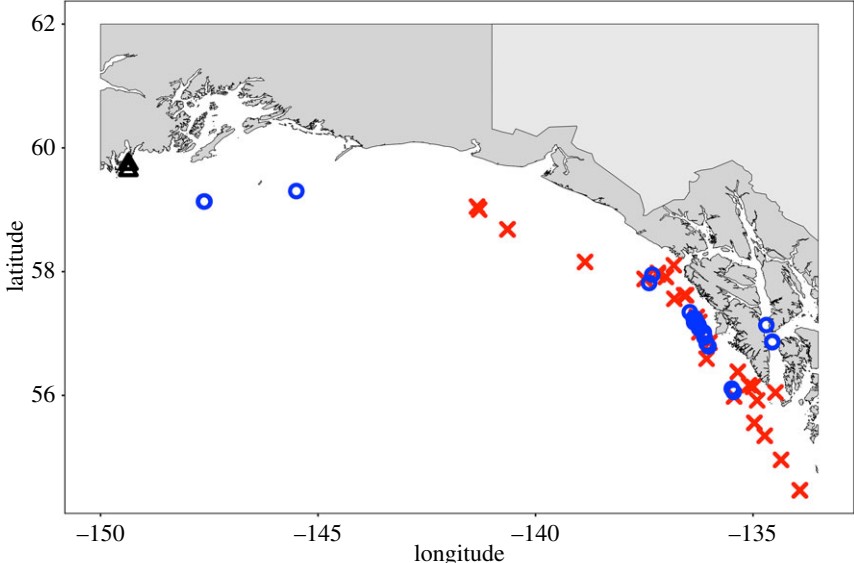

**Figure 1.** Map of the Gulf of Alaska showing locations where biopsy samples (blue circles) and prey (red x's) were collected. The two prey locations (black triangles) in the central GOA represent the locations of the three ragfish collected, all outside of the study area.

**Table 1.** Description of samples collected for this study: whether they were in historical stomach contents, collection year, gear type (BT = bottom trawl, LL = longline) and sample size.

| species | historical diet? | year | gear | *n* |
|---|---|---|---|---|
| predator | | | | |
| sperm whale | — | 2003–2017 | biopsy | 33 |
| prey | | | | |
| ragfish[a] | yes | 2017 | BT[b] | 3 |
| shortraker rockfish | yes | 2016–17 | LL[b] | 45 |
| skate | yes | 2016–17 | LL[b] | 42 |
| spiny dogfish | yes | 2016–17 | LL[b] | 34 |
| clubhook squid | yes | 2014–17 | LL[c] | 10 |
| magister squid | yes | 2016–17 | BT[b]/LL[c] | 42 |
| sablefish | no | 2016–17 | LL[b,c] | 45 |
| grenadier | no | 2016–17 | LL[b] | 44 |
| baseline | | | | |
| *Neocalanus* spp. | — | 2016–17 | bongo | 62 |

[a]Collected outside study area.
[b]Collected on NMFS survey gear.
[c]Collected from commercial fishermen.

Biopsies were collected at a distance of 15 m or greater, using a modified Barnett crossbow equipped with a 40 mm stainless steel tip [42]. Animal reactions were recorded before, during and after the samples were collected; all animals resumed pre-sample behaviour within one dive cycle after the samples were collected. The inner layer of skin (approx. 2 mm thickness) was used in this analysis, probably representing the most recent diet of whales [44].

## 2.2. Data collection—prey

Fish and squid species historically found in sperm whale stomachs, as well as other species bycaught by commercial longline gear and available to depredating sperm whales, were analysed for stable isotopes (figure 1 and table 1). Samples were collected and donated from a variety of sources, including

commercial longline fishermen fleet members, the NOAA GOA longline survey and the NOAA GOA bottom trawl survey (table 1). Fish identified in historical stomach contents studies were requested from these sources and included ragfish, rockfish, spiny dogfish and skates. Specific species of rockfish were not identified in whaling literature; therefore, we chose shortraker rockfish (*Sebastes borealis*) due to their large biomass relative to other rockfish species in the region and their prevalence in sablefish fishing habitat, suggesting they may be more available to whales from longline gear bycatch. Similarly, specific species of skates were not identified in whaling literature, so we requested all skate species bycaught on sablefish longline gear. All skate samples came from the NOAA GOA longline survey (table 1), and only longnose skate (*Raja rhina*) were identified to species by NMFS scientists collecting the samples; all other species were grouped as 'other' skates (specific species were not listed) though NMFS scientists collecting the samples noted that they consisted primarily of Alaskan and big skates. Robust clubhook squid (*O. robusta*) and magister armhook squid (*B. magister*) were reported in historical stomach contents and samples were requested and collected from commercial longline fishermen and the NOAA bottom trawl survey (table 1). Species that were not identified in historic stomach contents but were included due to their prevalence as bycatch on longline fishing gear and subsequent availability to sperm whales while depredating were giant grenadier (*Albatrossia pectoralis*) and Pacific grenadier (*Coryphaenoides acrolepis*). While difficult to observe direct feeding, sperm whales have been observed following closely behind longline vessels and abruptly dipping under the surface after grenadier have fallen from a hook near the surface of a longline haul (J Straley 2007, unpublished data).

We attempted to collect ragfish for this experiment due to their prevalence in the diets of sperm whales killed off the northeast coast of Vancouver Island and the eastern GOA during commercial whaling [8,11,12,14,15]. However, we were able to obtain only three specimens for this study, all of which were caught in shallow (less than 50 m) waters well outside of our study area (figure 1). Given the small sample size of three samples, none of which were collected in the study area, and coupled with a lack of any specimens found in habitat sperm whales are observed in or known to inhabit, we chose not to include them in the final analysis. However, we did run the full analysis with these three ragfish included, and it did not change our results (electronic supplementary material, appendix 1).

## 2.3. Data collection—baseline organisms

Isotopic baselines, typically defined as the isotopic signature of primary consumers, vary between ecosystems, can fluctuate over years, and are crucial to accurate trophic level calculations [32]. In the offshore pelagic environment calanoid copepods (*Neocalanus* sp.) have been used [32,45]. For this study, we collected and sorted *Neocalanus* spp. copepods from 200 m depth bongo tows (mesh size 505 μm) conducted offshore in the eastern GOA in July of 2016 and 2017 during NMFS GOA research surveys [46]. In 2016 and 2017, four and six stations were sampled, respectively, over the continental slope in water depths of 400–800 m. For each sample, copepods were sorted by species, with *Neocalanus plumchrus* and *Neocalanus flemegeri* having the highest abundances. Due to their similar size and life history, they were combined, and approximately 10 specimens were selected from each station for isotope analysis.

## 2.4. Stable isotope analysis

For each squid specimen, a 2–3 cm section of mantle and tentacle were sampled. For rockfish, dogfish and grenadier, a 2–3 cm section of dorsal muscle was sampled. Muscle samples from the neck or 'collar' area of sablefish were collected in order to retain the value of the fillet. A 2–3 cm section of skate muscle was sampled from the wing. Depth stratum (201–300, 301–400, 401–600, 601–800 and 801–1000 m), length and location (latitude, longitude) were recorded for each specimen, with the exception of clubhook squid and some magister squid, which were primarily collected from commercial fishermen who did not always record an exact location or depth. All samples were stored at −80°C until processing. All whale, squid and fish samples were first rinsed with de-ionized water, then cut into small pieces and oven-dried at 60°C for 24 h before being ground to a fine powder. Copepods were left whole. Lipids are known to be depleted in $^{13}C$ relative to $^{12}C$, causing tissues with high lipid content to show falsely decreased $\delta^{13}C$ values [35]. However, the same is not true for $\delta^{15}N$. Therefore, to account for differences in lipid content, samples were lipid-extracted prior to analysis. However, half of the sample was first saved to run separately for $\delta^{15}N$ values to alleviate any changes to $\delta^{15}N$ values caused by lipid extraction [47–51]. Lipid extraction was carried out using

three cycles in a 2 : 1 chloroform–methanol solution [44,52–54]. Samples were then again dried overnight, after which 0.2–0.4 mg aliquots were measured into tin capsules and sent to the Alaska stable isotope facility (ASIF) in Fairbanks, Alaska. Samples were processed at ASIF with an elemental analyser isotopic ratio mass spectrometer for bulk carbon and nitrogen, expressed in delta ($\delta$) notation, according to the equation:

$$\delta X(‰) = \left[ \frac{R_{sample}}{R_{standard}} - 1 \right] \times 1000, \tag{2.1}$$

where the isotope ratio X represents $^{13}$C or $^{15}$N, and $R$ represents the abundance ratio of each isotope ($^{13}$C/$^{12}$C or $^{15}$N/$^{14}$N). Duplicates were run on 15 sperm whales and on 10 of each prey item except clubhook squid, for which five duplicates were run. Stable isotope ratios are expressed in units of parts per thousand ('per mil', ‰). Reference standards used were Vienna Pee-Dee Belemnite for carbon and atmospheric $N_2$ for nitrogen. Analytical precision was ±0.2‰ and ±0.3‰ for $\delta^{13}$C and $\delta^{15}$N, respectively, calculated from replicates of peptone and duplicates.

## 2.5. Trophic enrichment factors

Stable isotope ratios of materials change as they are incorporated into a consumer's tissues; the difference in isotope values between predator and prey is known as a trophic enrichment factor (TEF), and can vary between species and between tissues within a single species [55]. Trophic level calculations, as well as dietary mixing models, require estimates of these rates. For marine mammals, estimation of TEFs can be complicated because direct control of feeding and/or continual observation of free-ranging marine mammals is nearly impossible. Thus many large whale species have estimated TEFs [56–58], while for pinnipeds and smaller cetaceans, some captive feeding experiments have been performed [59–61]. TEFs in the skin of free-ranging fin whales and pilot whales have been estimated for $\delta^{15}$N values at 2.8‰ and 1.7‰, respectively, and $\delta^{13}$C values at 1.3‰ and 1.2‰, respectively [56,57]. Two captive feeding experiments exist where groups of bottlenose dolphins were tested under different diets [59,60]. We combined raw data from these captive feeding experiments and calculated variance-weighted mean TEFs and their standard errors as inputs for mixing models (TEF = 2.12 ± 0.53‰ for $\delta^{15}$N; TEF = 0.96 ± 0.38‰ for $\delta^{13}$C). Given our estimates were roughly mid-way between the fin whale and pilot whale free-ranging estimates, we ran separate analyses using both the low (pilot whale) and high (fin whale) TEF estimates as a sensitivity analysis (electronic supplementary material, appendix 1). Dietary proportions were slightly different, but the relative importance of different prey items remain unchanged.

## 2.6. Objective 1: describe isotopic variability and calculate trophic position

### 2.6.1. Isotopic variability

The mean (±s.d.) and range of stable isotope ratios were calculated for sperm whales and all prey species. For prey that had been split into two species, such as the longnose skate and 'other' skates, as well as the Pacific grenadier and giant grenadier, isotopic ratios between the groups were explored using analysis of covariance (ANCOVA). This allowed us to explore how similar species were, and to determine whether or not the species should be grouped together for final dietary mixing model analysis or remain separated. Isotope ratios for each prey species were tested using ANCOVAs with respect to length and depth strata to assess whether prey should be split up in other ways from an isotopic perspective.

### 2.6.2. Trophic level calculation

The trophic level calculation is represented by

$$TL = 2 + \frac{(\delta^{15}N_{specimen} - \delta^{15}N_{primary\ consumer})}{2.12}, \tag{2.2}$$

where $\delta^{15}N_{specimen}$ represents the nitrogen isotope ratio values of the predator (i.e. sperm whales, fish, squid, etc.), $\delta^{15}N_{primary\ consumer}$ represents those of the baseline (calanoid copepods), 2 represents the assumed trophic position of the baseline consumer (*Neocalanus* sp.) and 2.12 represents the average enrichment per trophic level for whales. For trophic level calculations of fish and squid, a TEF of 3.5 was used [62].

**Table 2.** Stable isotope ratios ($\delta^{13}$C and $\delta^{15}$N values) and trophic level calculations of sperm whales and each of their potential prey items, ordered by trophic level.

| | n | $\delta^{15}$N ± s.d. (‰) | $\delta^{13}$C ± s.d. (‰) | trophic level |
|---|---|---|---|---|
| predator | | | | |
| sperm whale | 33 | 16.9 ± 0.8 | −17.2 ± 0.6 | 5.7 ± 0.4 |
| prey | | | | |
| clubhook squid | 10 | 16.7 ± 0.8 | −18.8 ± 0.6 | 5.7 ± 0.4 |
| skate | 45 | 16.1 ± 0.7 | −17.9 ± 0.5 | 5.4 ± 0.3 |
| shortraker rockfish | 45 | 15.1 ± 0.9 | −18.6 ± 0.5 | 4.9 ± 0.4 |
| sablefish | 45 | 14.4 ± 0.8 | −17.9 ± 0.6 | 4.6 ± 0.4 |
| grenadier | 45 | 14.1 ± 0.9 | −19.6 ± 0.6 | 4.5 ± 0.4 |
| spiny dogfish | 38 | 13.9 ± 0.9 | −17.8 ± 0.8 | 4.4 ± 0.4 |
| magister squid | 45 | 12.1 ± 1.1 | −18.8 ± 0.8 | 3.5 ± 0.2 |
| baseline | | | | |
| *Neocalanus* spp. | 62 | 8.9 ± 0.5 | −19.2 ± 0.9 | 2.0 |

## 2.7 Objective 2: assess sperm whale diet using isotopic mixing models

### 2.7.1. Dietary mixing models

To estimate proportional contributions of prey items to sperm whale diets, isotopic mixing models were used and implemented in the R packages MixSIAR (Bayesian Mixing Stable Isotope Analysis in R) and SIMMR (Stable Isotope Mixing Models in R) [38,63,64]. These models use Bayesian methods to account for uncertainty in input parameters, such as source isotope ratios and TEFs [37,38,64]. Isospace plots were used to examine how well the sperm whale samples (predator mixtures) fell within the isotopic space of the prey. Posterior probabilities were estimated using three chains of length 100 000 after a burn-in of 50 000 iterations and chains were thinned by subsampling every 50th iteration. All statistical analyses were performed using the free software package R v. 3.5.1 [65], and are available within the GitHub repository [66].

Models were run for all whales combined and for selected subsets of whales to test the following specific hypotheses:

*Hypothesis 1: Sperm whale diet composition did not change over the past 15 years.*

All whale samples were collected between 2003 and 2017 and were divided into 'older' samples collected more than 10 years ago (2003–2009), and 'recent' samples (2010–2017) to assess how dietary proportions may have changed over the past 15 years.

*Hypothesis 2: Sperm whale diet composition did not exhibit within-season changes.*

The field season each year ran from May to September, roughly considered 'summer', and samples were divided into early, mid and late summer time periods to assess potential within-season variability in diet.

*Hypothesis 3: Sperm whale diet composition was the same between frequent and non-frequent depredators.*

There is evidence that some whales in the SEASWAP catalogue are sighted more frequently than other whales. Of the 122 individual whales in the catalogue, 12 individuals make up one-third of all sightings and are referred to as 'frequent' depredators. Of the biopsy samples used in this analysis, 10 came from frequent depredators, while the remaining 23 were from non-frequent depredators. We assessed differences in dietary proportions between frequent depredators and non-frequent depredators.

## 3. Results

Thirty-three biopsy samples from sperm whales were archived by SEASWAP between 2003 and 2017 that had sufficient skin to include in stable isotope analysis. With the exception of ragfish ($n = 3$) and clubhook squid ($n = 10$), sample sizes for prey ranged from 38 to 52 samples (table 1). The clubhook squid were donated by fishermen and were caught on longline gear, with 10 samples being donated between 2014 and 2017 (table 2). Ragfish were requested from all NMFS surveys (NMFS longline survey and

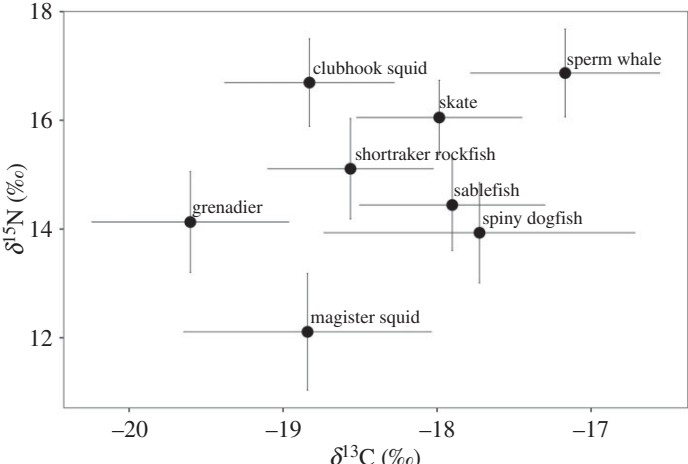

**Figure 2.** Stable isotope ratios of sperm whales and presumed prey items in the Gulf of Alaska. Points are means for each species, while error bars represent one standard deviation from the mean.

NMFS bottom trawl survey) as well as from the longline fleet members, but only three samples were caught between 2016 and 2017, all from water depths ≤50 m, which is shallower than known sperm whale habitat, and in a region outside of the study area.

## 3.1. Objective 1: describe isotopic variability and calculate trophic position

Mean stable isotope ratios of sperm whales ranged widely from −20‰ to −17‰ for $\delta^{13}$C and from 12‰ to 17‰ for $\delta^{15}$N (figure 2). $\delta^{15}$N values of sperm whales ranged from 15.2‰ to 18.3‰ ($\bar{x}$ ± s.d. = 17.0 ± 0.7‰), resulting in an estimated trophic level of 5.7 ± 0.4 (table 2 and figure 2). The estimated mean trophic levels of prey ranged from 3.5 for magister squid to 5.7 for clubhook squid, which fed almost at the same trophic level as sperm whales (table 2 and figure 2). Isotope ratios of the two skate groups (longnose skate versus 'other' skates) were not significantly different (ANCOVA: $p > 0.14$; table 3). Similarly, there were no significant differences in either $\delta^{13}$C or $\delta^{15}$N between Pacific grenadier and giant grenadier (ANCOVA: $p > 0.16$; table 3). Therefore, all skate species and both grenadier species were grouped as 'skates' and 'grenadier', respectively. The $\delta^{13}$C and $\delta^{15}$N of magister squid and the $\delta^{13}$C values of spiny dogfish differed significantly among depth strata (table 3). Nevertheless, because isotope ratios of each depth strata overlapped so closely, and all depth strata were within sperm whale habitat, we considered all specimens of a given species to be representative of the available prey and combined them into a single group across depth strata for use in mixing models. This was necessary to reduce the number of end members, and thus uncertainty in the model. Similarly, $\delta^{13}$C values of grenadier increased significantly with length (ANCOVA: $F_{30,38} = 23.49$, $p < 0.001$), but all grenadier were combined into a single group for the mixing model because they had a narrow length range (25–40 cm for pre-anal fin length), were found in the same habitat, and were assumed to be representative of grenadier that sperm whales might consume. This was also done to reduce end members and uncertainty in the models and group prey where possible. Finally, $\delta^{15}$N values of magister squid increased significantly with length (ANCOVA; $F_{36,42} = 63.30$, $p < 0.001$). All length classes were again grouped for the reasons noted above. In addition, isotope ratios were more similar among conspecifics than among species and were thus naturally grouped by species.

## 3.2. Objective 2: assess sperm whale diet using isotopic mixing models

Sablefish and spiny dogfish had very similar isotopic ratios (figure 2 and table 2), indicating they probably occupy a similar trophic niche space and would be difficult to differentiate in mixing models. Consequently, we grouped these two species together for dietary mixing model analysis. We also ran models with the two species separated, and patterns of dietary proportions between prey items did not change, but those for dogfish and sablefish increased when they were combined (see electronic supplementary material, appendix 1).

**Table 3.** Analysis of co-variance (ANCOVA) results for each species and isotope relationships with length and depth. For skates and grenadier, species level relationships were tested as well. Results in italics are those that were significant. Clubhook squid depth strata and lengths were not consistently or accurately recorded by commercial fishermen so they were not included.

| species | factor | $\delta^{13}$C | | $\delta^{15}$N | |
|---|---|---|---|---|---|
| | | F-value | p-value | F-value | p-value |
| clubhook squid | length | | | | |
| | depth.strata | | | | |
| skate | length | $F_{42,43} = 34.90$ | 0.001 | $F_{42,43} = 3.12$ | 0.128 |
| | depth.strata | $F_{5,43} = 5.15$ | 0.052 | $F_{5,43} = 5.25$ | 0.048 |
| | sub.species | $F_{1,43} = 0.23$ | 0.648 | $F_{1,43} = 2.78$ | 0.146 |
| shortraker rockfish | length | $F_{25,38} = 0.05$ | 0.817 | $F_{25,38} = 0.58$ | 0.449 |
| | depth.strata | $F_{5,38} = 0.71$ | 0.556 | $F_{5,38} = 1.29$ | 0.291 |
| sablefish | length | $F_{26,32} = 0.78$ | 0.385 | $F_{26,32} = 3.85$ | 0.058 |
| | depth.strata | $F_{5,32} = 1.21$ | 0.326 | $F_{5,32} = 0.44$ | 0.816 |
| grenadier | length | $F_{30,38} = 23.49$ | <0.001 | $F_{30,38} = 3.65$ | 0.064 |
| | depth.strata | $F_{5,38} = 0.10$ | 0.961 | $F_{5,38} = 0.38$ | 0.767 |
| | sub.species | $F_{1,38} = 2.01$ | 0.164 | $F_{1,38} = 1.55$ | 0.221 |
| spiny dogfish | length | $F_{26,32} = 0.81$ | 0.375 | $F_{26,32} = 4.62$ | 0.041 |
| | depth.strata | $F_{5,32} = 4.84$ | 0.007 | $F_{5,32} = 1.58$ | 0.214 |
| magister squid | length | $F_{36,42} = 3.61$ | 0.072 | $F_{36,42} = 63.3$ | <0.001 |
| | depth.strata | $F_{5,42} = 9.21$ | <0.001 | $F_{5,42} = 6.40$ | <0.001 |

Isotope ratios of predator and prey samples plotted together, called an 'iso-space plot', generally showed sperm whale samples (mixtures) to lie within the mixing polygon defined by the mean of the sources, indicating the data were acceptable for running the mixing models (figure 3) [37]. Five mixtures branched off the main cluster with decreased $\delta^{13}$C and $\delta^{15}$N values (figure 3); however, they did not come from a specific year, month or region that would separate them out from the other mixtures in this analysis.

Mixing models for all sperm whale samples and whale groupings showed the sablefish/dogfish group, skates and rockfish to be the major dietary contributors (table 4). The estimated proportions were highly variable, reflecting variability in diet composition as well as a high degree of uncertainty. The sablefish/dogfish group and skates were the largest contributors to all whale diets in general (mean ± s.d.: 35.6 ± 13.9% and 25.4 ± 8.3%, respectively). All mixing model results suggest that grenadier, clubhook squid and magister squid generally made up a smaller proportion of sperm whale diets, with each species contributing less than 10% to diets of all whale groups except magister squid in the mid-summer whale group (12.7 ± 10.2%; table 4).

*Hypothesis 1: Sperm whale diet composition did not change over the past 15 years.*

The results of mixing models show that the proportion of sablefish/dogfish in diets was higher in recent samples (58.1 ± 22.5%) than older samples (34.7 ± 13.3%). Similarly, the proportion of shortraker rockfish increased from older samples (9.9 ± 4.7%) to more recent samples (21.5 ± 20.3%) (table 4 and figure 4). Model outputs were used to compute the probability that more recently sampled whales had a higher proportion of sablefish/dogfish in their diets than older samples, with a probability of 86%.

In exploring the isotopic niche of sampled whales over time, a multivariate analysis of variance (MANOVA) indicated significant differences in $\delta^{13}$C and $\delta^{15}$N values among years ($F_{16,48} = 3.23$, $p = 0.001$). Subsequent univariate tests for each isotope ratio showed $\delta^{13}$C values to vary significantly at the 5% level ($F = 2.98$, $p = 0.02$), while $\delta^{15}$N values did not vary significantly by year ($F = 2.06$, $p = 0.08$).

*Hypothesis 2: Sperm whale diet composition did not exhibit seasonal changes.*

Seasonal mixing model results showed the contribution of sablefish/dogfish increased seasonally from 25.8 ± 20.2% in early summer samples to 43.9 ± 19.5% in late summer samples. Skates contributed the largest proportion to early summer sperm whale diets (30.6 ± 20.3%) (table 4 and figure 4).

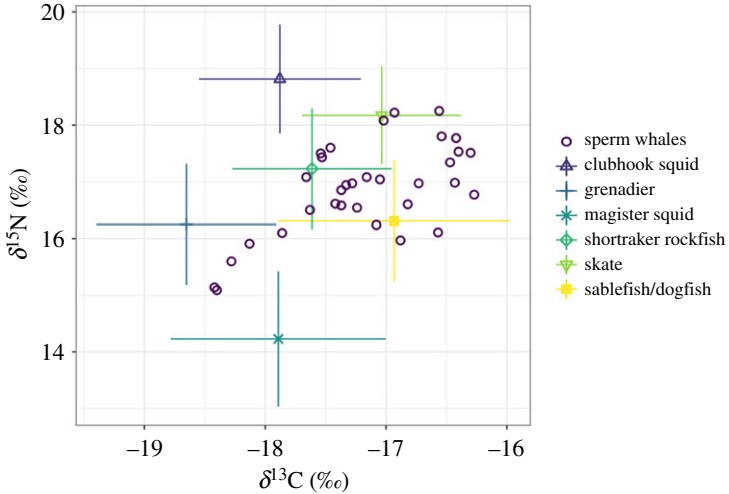

**Figure 3.** Isospace plot showing how sperm whale samples (mixtures) fit within prey space after trophic enrichment factors have been applied to prey. Error bars indicate combined source and discrimination uncertainty ±1 s.d.

*Hypothesis 3: Sperm whale diet composition was the same between frequent and non-frequent depredators.*

When frequent and non-frequent depredator groups were compared in mixing models, the diets of frequent depredators were dominated by skates (46.0 ± 20.4%), while non-frequent depredator diets were dominated by the sablefish/dogfish group (48.5 ± 22.3%) (table 4 and figure 3). From model outputs, the probability that the proportion of skates in the frequent depredator diets was higher than that of non-frequent depredator diets was 82%. Similarly, the probability that the proportion of sablefish/dogfish in the non-frequent depredator diets was higher than that of the frequent depredator diets was 84%.

# 4. Discussion

This work highlights trophic connections among sperm whales, groundfish and squid in the GOA, and identifies important prey species that contribute to sperm whale diets in this region. Sperm whales are a top predator in this ecosystem (TL = 5.7), closely followed by clubhook squid (TL = 5.7), while magister squid had the lowest trophic level estimates (TL = 3.5, table 2). Grenadier had $\delta^{13}$C values more negative than expected, given they generally inhabit the same areas as most other groundfish species sampled (figure 2). This could indicate they spend more time farther offshore and/or foraging in more pelagic habitat than the other species.

## 4.1. Sperm whale diet

In general, our mixing model results suggest that sperm whales sampled in this study area primarily consumed sablefish/dogfish, skates and shortraker rockfish (table 4). The apparent prevalence of skates, rockfish and dogfish is consistent with historical stomach contents data from commercial whaling in the mid-1900s, but the notable lack of ragfish and the addition of sablefish distinguishes more recent diets. Whales sampled in the early 2000s had a smaller proportion of the sablefish/dogfish group in their diets than whales sampled more recently between 2009 and 2017, indicating a potential increase in sablefish consumption over time. This complements fishermen's observations that depredation has been increasing over time. MANOVAs indicated year was a significant predictor in $\delta^{13}$C values of whales ($F_{16,48} = 3.23$, $p = 0.001$), though we must note the small sample sizes of one to six samples in all but one year which could cause individual variability to bias the results of these tests. Nevertheless, our results indicate there does seem to be fluctuation in the isotope ratios, and potentially diet, of whales across years. Larger sample sizes in future years would help tease out these relationships.

Sperm whales appear to increase sablefish/dogfish consumption throughout the summer fishing season. We note that inferring the time period of prey consumption that is reflected in a given sample is still highly uncertain for whales. Various studies on isotopic incorporation rates of blue whales and bottlenose dolphins have estimated full turnover of skin tissue between 163 ± 91 and 180 ± 71 days,

**Table 4.** Summary of estimated contributions (mean ± s.d.) of each prey item to sperm whale diets. Whales are grouped by all whales, older/recent samples, early/mid/late summer samples and frequent/ non-frequent depredators.

| species | all | temporal | | seasonal | | | depredation activity | |
|---|---|---|---|---|---|---|---|---|
| | | old | recent | early | mid | late | frequent | non-frequent |
| clubhook squid | 8.3 ± 6.6 | 5.4 ± 6.8 | 7.8 ± 7.1 | 7.2 ± 9.6 | 6.7 ± 7.2 | 5.6 ± 6.5 | 7.6 ± 6.1 | 5.9 ± 4.3 |
| skate | 25.4 ± 8.3 | 18.1 ± 7.7 | 21.5 ± 12.4 | 30.6 ± 24.3 | 17.7 ± 10.3 | 26.2 ± 14.9 | 46.0 ± 20.4 | 18.4 ± 15.7 |
| shortraker rockfish | 14.5 ± 11.1 | 9.9 ± 4.7 | 21.5 ± 20.3 | 25.8 ± 21.9 | 28.2 ± 21.9 | 11.9 ± 10.2 | 11.6 ± 11.5 | 15.4 ± 12.6 |
| sablefish/dogfish | 35.6 ± 13.9 | 34.7 ± 13.3 | 58.1 ± 22.5 | 25.8 ± 20.2 | 27.5 ± 17.8 | 43.9 ± 19.5 | 25.1 ± 15.1 | 48.5 ± 22.3 |
| grenadier | 5.7 ± 4.7 | 3.3 ± 4.1 | 5.3 ± 6.3 | 3.7 ± 4.5 | 7.2 ± 6.9 | 3.8 ± 3.4 | 3.4 ± 4.7 | 3.6 ± 4.9 |
| magister squid | 10.6 ± 6.8 | 5.4 ± 3.8 | 9.1 ± 7.2 | 6.7 ± 6.8 | 12.7 ± 10.2 | 8.7 ± 7.9 | 6.2 ± 6.1 | 8.1 ± 9.5 |

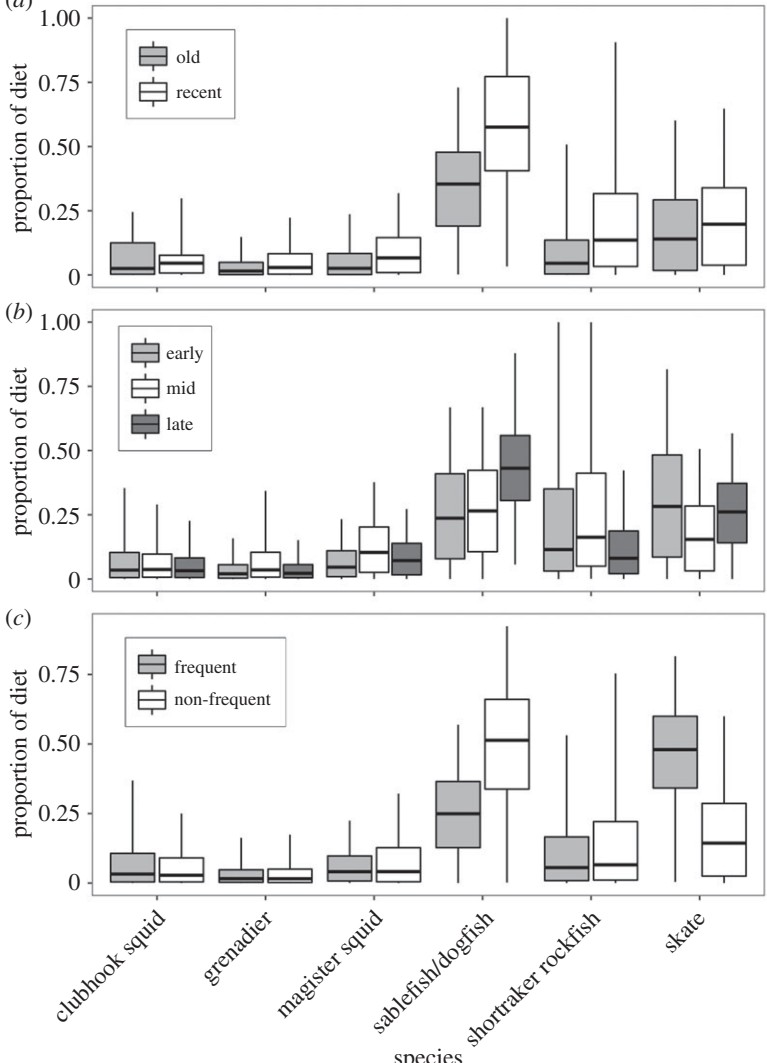

**Figure 4.** Boxplots showing mixing model estimates of the proportional contribution of each prey to sperm whale diets to compare (*a*) older (2003–2009) versus more recent (2010–2017) samples; (*b*) early, mid and late summer samples; and (*c*) frequent versus non-frequent depredators. Boxes represent lower and upper quartiles with a median line, while ends of whiskers show 95% credible intervals.

respectively, for $\delta^{15}$N values [58,60], which would result in the inner layer of skin representing diet from up to approximately 60 days prior to sampling [44]. If we apply this rate to our sperm whale samples, the early summer samples (mid-May through June) would represent sperm whale diet from mid-March to May, which aligns with the typical mid-March opening of the commercial longline fishing season each year. Conversely, the late summer samples collected from mid-August to September probably represent diet from mid-June through August. Therefore, these results indicate that the proportion of sablefish in sperm whale diets increases from the start of the longline fishing season through the summer.

The most surprising results from this study were that whales that are sighted most frequently in the SEASWAP catalogue consumed a higher proportion of skates than any other prey, while non-frequently sighted depredators had a larger proportion of the sablefish/dogfish group in their diets (table 4 and figure 4). While our sample size for the frequent depredators was low ($n = 10$), it is difficult to explain these findings. Biomass estimates for skates are similar to spiny dogfish, at approximately 50 000 t [67], which is much lower than sablefish biomass estimates (approx. 488 000 t) [68]. Further, the bycatch of skates is typically higher on halibut fishing sets than sablefish sets, indicating they are found at shallower depths than sablefish. Catch data corroborate that skates on average inhabit shallower depths than sablefish, though their habitats do overlap [67]. Therefore, while skates are bycaught on sablefish fishing gear, they are unlikely to be more available to frequently depredating sperm whales than sablefish. Calorimetry analysis of the lipid content available in each of these species may shed additional light on the values of skates to sperm whales but was outside the scope of this study.

One potential explanation for the high proportion of skates in frequent depredator diets is that a diet item was missing in the analysis that occupies the same isotopic space as skates. However, we are confident that we sampled the major available prey species in the habitat where sperm whales are foraging, especially depredating sperm whales. Species we did not sample were other species of rockfish, such as shortspine thornyheads (*Sebastolobus alascanus*), which are another large bycatch fish in longline fisheries, and Pacific ocean perch (*Sebastes alutus*), both of which were occasionally noted in sperm whale stomachs, though rarely [8,13]. While specific isotopic estimates for these species in GOA waters do not exist, trophic level estimates indicate that isotope ratios are probably very similar to the shortraker rockfish we sampled [69]. Therefore, it is unlikely either of these species occupy the same isotopic space as skates and were, therefore, an important diet item that was missed in this analysis.

Another explanation for the high proportion of skates in the diets of frequent depredators is that these animals actually prefer skates, and the frequency with which they follow longline fishing vessels brings them to shallower waters where fishermen often deploy 'combo sets'. These combo sets target both halibut and sablefish. On these shallow sets, skates may be relatively more abundant and may be targeted in addition to sablefish by frequent depredators.

Additionally, it is possible that while frequent depredators are targeting sablefish and skates on longline gear while depredating, they may also target skates while naturally foraging. Indeed stomach contents data from sperm whales killed off central California shows that after clubhook squid, longnose skates were the second-highest diet item [70].

The frequent depredator samples were collected across the summer months (early, mid and late summer samples), though two-thirds of them were collected during July, which would reflect the diet of whales back to May. A few fishermen have noted that in certain areas of the GOA they catch a lot of skate egg sacks on their longline gear while fishing for sablefish in May and early June. This may suggest that skates move into deeper water nurseries in early summer, where depredation and whale activity is more prevalent. Indeed there were reports of skate eggs found in sperm whale stomachs during commercial whaling, though the report came from the western North Pacific, near Olyotorsky Bay on the Russian coast [9]. In our study, early-season diets of frequent depredators were based on six samples that may be driving the apparent skate preference if these whales consumed a large proportion of skates while foraging and depredating around skate nurseries at that time of year.

While our plots generally showed prey covering our mixtures (sperm whale samples) sufficiently, there were a small number of mixtures that were slightly outside of the prey isospace when drawing a polygon around source means (figure 3) [37]. This is an indication that a potential prey item could have been missing, or TEFs could be underestimated. Our TEFs were estimated from literature values, and models were run with both a low and high estimate from marine mammal studies (electronic supplementary material, appendix 1). While the low and high TEFs resulted in slightly different isospace plots, general patterns were the same, and the same group of mixtures remained outside of the prey space. The other potential prey items from their GOA foraging grounds such as Pacific halibut or other rockfish species probably do not occupy isotopic space where the outlying mixtures were located, as discussed above. Some fishermen have wondered if sperm whales eat Pacific sleeper shark (*Somniosus pacificus*) in the region, though no historic evidence exists to support this theory, nor were we able to find isotope values to include in or compare with our study.

An alternative explanation for small differences between mixtures and prey in isospace plots is that there were prey items consumed by sperm whales outside of Alaskan waters. The samples that fall near the outer edges of the sampled prey in figure 3 could have come from whales that had recently arrived in the GOA and were feeding on other species on their way north. Humboldt squid (*Dosidicus gigas*) are a preferred prey item in warmer equatorial waters off Mexico and Central America [33,71]. SEASWAP has tracked a number of sperm whales using satellite tags, which have made broad-scale movements from GOA waters to Mexican waters [72]. Isotope studies that have been done in these areas have found delta values of *D. gigas* consistent with where the outlier mixtures were located in the isospace plots, after applying TEF estimates (figure 3) [33,71]. Recent range expansions of *D. gigas* into Northeast Pacific waters further suggests the GOA sperm whales sampled in this study may have been feeding on these species as they moved toward the GOA [73].

## 4.2. Caveats

While ragfish were listed as an important diet item to sperm whales in our study area during historical commercial whaling, we were unable to collect any specimens in our study area during this study. This could indicate that ragfish are less abundant than they were historically or that they primarily occupy a

different habitat to those we were able to sample through surveys and opportunistic fishery donations, such as higher in the water column or at deeper depths. Indeed, there is some evidence from descriptions of ragfish that they are adapted to inhabit deep depths [14]. It is also possible they are not vulnerable to the gear used to collect samples (bottom trawl and demersal longline gear). From an isotopic perspective, if they lived much deeper, or in a different place in the water column, we would expect their isotope ratios to reflect that different habitat [66,69,74–76]. However, the isotopic composition of the three specimens we were able to acquire to the northwest of our study region were similar to those of spiny dogfish, indicating they probably inhabit a similar depth range or have similar prey preferences (electronic supplementary material, appendix 1).

The isotopic similarities between sablefish and dogfish were unfortunate in that sablefish were the species of interest in this study. Mixing models would be unable to differentiate between the two sources, and thus they were grouped together. Both species are demersal fish as adults, found in the same general habitat of the continental slope, and stomach contents research has shown a majority of their diets to both be from fishery offal, though their diets do differ slightly between crustaceans and fishes [77,78]. This suggests they may occupy different trophic niches, though similar isotopic niches. We believe that whales feeding in the isotopic space of these two sources are probably consuming much larger proportions of sablefish than dogfish for several reasons. First, recent sablefish biomass estimates (approx. 488 000 t) [68] are more than triple that of spiny dogfish (55 000 t) [79] in the GOA, indicating they are probably more biologically available. Second, sablefish are targeted by sperm whales in depredation activity [6,17,18]. Therefore, while the group may include some spiny dogfish, a majority of the group probably comprises sablefish. Future analyses using fatty acid signatures of prey and sperm whale tissue may be able to estimate the proportions of sablefish and dogfish in sperm whale diets.

It is important to note that this study was done using data that was often collected opportunistically during other projects. As such, samples were collected only during summer months (May to September), and in sometimes only during a single month or week of a given year. No samples were collected outside of longline season during the winter months, when sperm whales may be naturally foraging in GOA waters. Peak effort for longline fishing typically happens in the first few months of the fishery in March and April, and fishermen often try to fish as early in the season as possible when they believe there are fewer sperm whales on the GOA fishing grounds. This probably has an impact on how depredation is spread out throughout the fishery, and how prey preferences and dietary contributions shift throughout the season. Future work to solidify isotopic incorporation rates, coupled with more dedicated sampling effort through all seasons of the year, would help elucidate some of the remaining questions regarding sperm whale prey preferences in the GOA and how they are changing over time.

## 4.3. Summary and next steps

This work represents an important first step in describing the foraging ecology of sperm whales and their prey in the GOA. While some work has been done to explore diving behaviour and how whales interact acoustically with longline fishing gear in this region [25], this project highlights patterns in the contribution of various prey to sperm whale diets. These findings have implications on the management of groundfish stocks and provide specific insights to managers on the commercially valuable sablefish stock. The lack of sablefish as sperm whale diet in historical stomach contents records from commercial whaling suggests that consuming sablefish is relatively new for sperm whales and that depredation reflects a new source of mortality that sablefish did not experience in the past. Hence this mortality should be accounted for in the stock assessment, separately from other natural mortality. While sablefish were not an important prey item historically, they currently make up over 50% of the diet for individuals in this region, which is due at least in part to depredation activity. This apparent continued increase in the proportion of sablefish in diets is noteworthy and is consistent with looking at the same issue from the NMFS survey and fishermen perspectives, which have also shown a potential increase in depredation-related mortality over time and a behavioural spreading of depredation [21,80,81]. Our work suggests that sperm whales have not only changed behaviours to target sablefish on longlines, but that they actually switched their prey in response to discovering the easy availability of the lipid-rich sablefish food source. Future work to compare the energy content of all prey items in this study may inform whether or not whales have switched to a better prey resource in sablefish, and more importantly speak to ecosystem functions and how changes in trophic pathways may be impacting energy flow and ecosystem functions.

While the recovery of whales after the secession of commercial whaling is often cited as an increase in resource conflicts throughout the world, this work shows that at least for sperm whales, the conflict could simply have arisen from whales adopting a new diet in response to an opportunity, regardless of the changes in the number of sperm whales. Working with the commercial fleet and managers together through collaborative research projects will be an integral part of further understanding how to manage sperm whale interactions with both fisheries and the primary prey species (sablefish, skates and rockfish) identified in this project.

Ethics. All sperm whale tissue samples in this study were granted to Jan Straley and Lauren Wild under NOAA Fisheries Research Permit nos. 14122 and 18529 and Institutional Animal Care and Use Committee (IACUC) permit no. 906340-4. These permits covered all fieldwork carried out for this study. All fish and squid samples were either donated by the NOAA Fisheries Gulf of Alaska Longline Survey's commercially harvested catch, the NOAA Fisheries Gulf of Alaska Bottom Trawl Survey's commercially harvested catch or donated by commercial fishermen after being harvested commercially.

Data accessibility. All of the data used in this work can be found in the Dryad Digital Repository: https://doi.org/10.5061/dryad.8th871g [82]. Relevant code for this research work are stored in GitHub: https://github.com/LaurenWild/UAF_SpermWhale_Diet and have been archived within the Zenodo repository: https://zenodo.org/record/3633440#.XjTSk1NKjfY.

Authors' contributions. L.A.W. participated in the study design, collected field data, carried out the laboratory work, participated in data analysis and drafted the manuscript; J.M.S. participated in the study design; assisted with historical records compilation, and critically revised the manuscript; B.W. and F.M. participated in study design, assisted with data analysis, helped draft the manuscript and critically revised the manuscript. All authors gave final approval for publication and agree to be held accountable for the work performed therein.

Competing interests. We declare we have no competing interests.

Funding. The work for this project was funded through a graduate mentoring research assistantship (GMRA) as part of the BLaST Program at the University of Alaska Fairbanks. The research reported in this publication was supported by the National Institute of General Medical Sciences of the National Institutes of Health under award nos. UL1GM118991, TL4GM118992 or RL5GM118990. The content is solely the responsibility of the authors and does not necessarily represent the official views of the National Institutes of Health. Funding for historical sample collection came from the North Pacific Research Board (award nos. 0309, 0412, 0626, 0918 and 1217), NOAA's Saltonstall-Kennedy program (award no. NA15NMF4270271), Alaska Longline Fishermen's Association and Central Bering Sea Fishermen's Association.

Acknowledgements. Data were collected in collaboration with Cascadia Research Collective, Scripps Institution of Oceanography, Alaska Sea Life Center, Alaska Longline Fishermen's Association and the Sitka Sound Science Center. SEASWAP co-PIs were all integral in making this project happen: Linda Behnken, Dan Falvey, Victoria O'Connell, Aaron Thode and Russ Andrews. John Calambokidis and Greg Schorr collected biopsy samples used in this project. Kelly Robertson and Gabriela Serra-Valente archived samples at Southwest Fisheries Science Center. Special thanks to the commercial longline fishermen who donated fish and squid that they caught: Frank Balovich and Cale Laduke (F/V Carole D), Paul Ipok (F/V Myra), Walt Cunningham and Jeff Farvour (F/V Christi-Rob), Ryan Nichols (F/V Nekton), Stephen Rhoads and Nick Nekeferof (F/V Magia), Phil Wyman and Kevin Johnson (F/V Archangel), Lucas Skordahl (F/V Tyee), Tyrus Moffitt and Alek Dyakanoff. NMFS GOA longline survey, bottom trawl survey and ecosystem assessment cruise personnel collected specimens: Chris Lunsford, Cindy Tribuzio, Pete Hulson, Dana Hanselman, Cheryl Barnes, Nancy Roberson, Jamal Moss and Wes Strasburger. Laboratory and analysis assistance provided by Illiana Ruiz-Cooley, Todd Miller, Casey Clark, John Logan, Andrew Parnell, Ellen Chenoweth, Madison Kosma, Mike Sigler, Corey Fugate, Matt Rogers, Kate Hauch, Michelle Parke, Kristina Long, Nevé Baker, Emily Whitney and Annie Masterman. Jen Cedarleaf archived historical samples and managed the database. The Inter-Library-Loan folks with the UAF Rasmussen library found all kinds of crazy whaling documents. Finally, special thanks to the Alaska Stable Isotope Facility team of Mat Wooller, Tim Howe and Norma Haubenstock for their work running bulk isotopes for all of these samples.

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
