## [Reviewer comments · Royal Society Open Science]

Review History

RSOS-191110.R0 (Original submission)

Review form: Reviewer 1

Is the manuscript scientifically sound in its present form?

No

Are the interpretations and conclusions justified by the results?

Yes

Is the language acceptable?

Yes

Do you have any ethical concerns with this paper?

No

Have you any concerns about statistical analyses in this paper?

No

Recommendation?

Major revision is needed (please make suggestions in comments)

Comments to the Author(s)

The manuscript titled “Exploring variability in the diet of depredating sperm whales in the Gulf of Alaska through stable isotope analysis” uses stable isotope analysis to identify a trophic shift in sperm whales. Over the past decades they have shifted their diet to a high dependence on Sablefish, and they achieve this by depredating catches from GOA long line ground fisheries. Overall, this paper is well-written and shows understanding of the caveats associated with stable isotope analysis and mixing models. The authors have done a good job at addressing these where possible. The biggest concern is the TEFs used, as these are estimated from other species. TEFs are affected by so many variables, it is hard to be sure that the TEFs between these species are comparable. However, after a short literature review I concluded that the authors have done their best to find the most appropriate values and that those used in the manuscript are the best available options.

Nevertheless, I have identified a few points that need to be addressed before this paper is acceptable for publication.

Specific comments:

Throughout manuscript: Line numbers don't match lines

Page 1 Line 22: “157,680 sperm whales”, put this into context. What is the current population, what was the estimated population then? I don't know whether to think this is a lot or not.

Page 2 Line 17: This does not need to be added to the manuscript, but I am curious, how can one tell that a whale is picking up vessel-hauling sounds from 10 miles away?

Page 3 Line 31: You mention the downsides to using fecal samples and direct observation to assess diet. Despite the low sample sizes, these studies may be the only direct insight we have to diet in recent years. You based the included prey types in your models off of stomach contents from decades ago, how do those data align with more recent studies?

Page 4 Line 8: A more recent reference: Barton et al. 2017

Page 5 Line 24: How similar?

Page 6 Line 43: Species should not be included in mixing models unless they have been observed in diets. Have sperm whales been observed feeding on these grenadier species? If not, they should be removed from the analysis.

Page 7 Line 48: What is the difference between a 1 inch chunk and a 1 inch section? This should also be in centimeters.

Page 8 Line 50: What standards were used to determine analytical precision? Were duplicate samples used to assess sample homogeneity? What reference materials were used?

Page 9 Line 8: This sentence is awkward, and grammatically incorrect. Try “Stable isotope ratios of materials change as they are incorporated into a consumer's tissues; the difference in isotope values between predator and prey is known as a Trophic Enrichment Factor (TEF).” No need to mention isotopic incorporation rates as you don't use that terminology throughout the paper.

Page 9 Line 12: Remove “primarily for $\delta^{15}\text{N}$ values”

Page 9 Line 24: Are there no estimations for carbon TEFs?

Page 10: All page numbers from here on out appear as “1” from 10-19 and “2” from 20-29, and “3” onwards.

Page 12 Line 25: What TEFs were used to calculate trophic levels of non-sperm whale species?

Page 12 Line 34: Why did you use ANCOVA for skates and ANOVA for Grenadier?

Page 13 Line 20: Isotopic ratios and Isotopic space are the same thing, instead of isotopic space I believe you mean “trophic niche space”

Page 14 Line 35: Is this supposed to say “more positive”? A more negative value suggests feeding closer to shore.

Page 16 Line 43: One possible explanation that isn't addressed is energetic requirements. Are the more frequent depredators larger? Fatter? Healthier than the less frequent depredators? Perhaps the more frequent ones are seen more frequently because they spend more time foraging than the others, and therefore when longlines are not available they continue to feed on skates.

Page 18 Line 6: remove “in”

Page 21 Line 12: The lipid rich aspect is an interesting one. If the data is available, would it be possible to compare the lipid content/energy content of the prey items. This could suggest whether these whales have shifted to a better prey resource, but more importantly speaks to

ecosystem function rather than just food web structure. If sablefish are high energy fish, and this new trophic pathway is changing energy flow pathways so that it is bypassing other dependent predators, this could have significant impacts on ecosystem functions.

Figure 3: The 5 mixtures that branch off of the main cluster that have decreased N and C values, were those from any particular region? Looking at the sampling map it appears as though approximately 5 whale samples were taken in different locations than the rest, and I wondered if these were those samples. If so, I would be interested to see if predicted dietary contributions would change if these outliers were removed.

Review form: Reviewer 2

Is the manuscript scientifically sound in its present form?

Yes

Are the interpretations and conclusions justified by the results?

Yes

Is the language acceptable?

Yes

Do you have any ethical concerns with this paper?

No

Have you any concerns about statistical analyses in this paper?

No

Recommendation?

Accept with minor revision (please list in comments)

Comments to the Author(s)

This is very interesting paper which does a good job at demonstrating the impact that disappearing prey and increasing fisheries has had on sperm whale diets and really comes together in the discussion. After some improvements, corrections and edits, I believe this paper will be of great value to our understanding of fisheries' impacts on the ecology of marine organisms and ecosystems.

I would suggest reorganising the introduction, notably by switching sections 1.2 and 1.3 around to provide a more streamlined understanding of the study subject. Start with the general theme of sperm whale diet and history and narrow down to the subject of depredation which will aid the flow leading to the goals described in 1.5.

Page 8, line 24: Two samples from whales were taken outside of the study area but included due to similar isotope values. However, Page 10, line 8, specimens of prey sampled outside of the study area were not included in the final analysis. Inconsistency of methodology?

Page 9, Line 29: Two groups of skates are chosen in the analysis: "longnose skates" and "other skates".

1. It is stated that "other" skates are most likely to be Alaskan and Big skates (Page 9, line 33), what is the evidence for this?

2. These samples come from fisheries' bycatch, were they identified by fishermen? If so, fishermen are likely to misidentify and mislabel skates which are notoriously prone to confusion (Iglesias et al. 2009; Dulvy et al. 2000). It is likely that among "other skates" were also counted longnose skates, in that case their isotopic ratios may have skewed the "other" group's isotopic ratio towards non-significance. It would be interesting and pertinent to see that variability in isotope ratios within the group does not suggest different species with different ratios and thus different contributions to the sperm whales' diets.

Page 12, line 31: "While bottlenose dolphins are much smaller than sperm whales, they are a toothed cetacean that consumes fish as prey that may have trophic levels comparable to sperm whale prey." This is a highly speculative statement, what is the evidence for their trophic levels being comparable?

Page 14 hypothesis 1: A major objective of the paper is to question whether the diet of sperm whales has changed over the past 15 years, but the results don't present an analysis of the trophic level of sperm whales between years. If possible, I would be interested in seeing a year to year comparison of the isotope data to present any gradual change over time of sperm whale diet.

Page 14, Hypothesis 2: The hypothesis questions seasonal changes in sperm whale diet, however sampling is only carried out over 1 season, how can seasonal variability be assessed? I would suggest reformulating the hypothesis to state that the variability considered here is within a season and not among seasons.

Page 14, line 34: some whale individuals are classed as "frequent" depredators. The methodology does not indicate how depredation is assessed, besides being sighted in proximity to longlines, what is the guarantee that these whales are actually depredating on lines?

Page 15, line 15: I would recommend changing the titles of subsections from "Objective x" to the actual objective description for ease of reading.

Page 15, line 42: "The $\delta^{13}\text{C}$ and $\delta^{15}\text{N}$ of magister squid and the $\delta^{13}\text{C}$ values of spiny dogfish differed significantly among depth strata (Table 3). Nevertheless, because all depth strata were within sperm whale habitat, we considered all specimens of a given species to be representative of the available prey and combined them into a single group across depth strata for use in mixing models."

All depths strata were combined into a single group because they were all within sperm whale habitat. But with if prey item isotope values were significantly different by depth strata, it would be interesting to see an analysis by depth strata to understand if whales are feeding at specific depths and how this might have change over the past 15 years with the onset of depredation.

Page 18, line 20-40: Prey items were grouped due to their similar isotopic niche. Justification was given for sablefish likely being the largest contributor of the 2 preys. However, a similar isotopic niche does not equate to a similar trophic or environmental niche. Does our knowledge of these prey items' ecology and that of sperm whales allow us to infer which prey may be contributing most? i.e. habitat use of sperm whales vs habitat use of prey. We know that sablefish are depredated on by whales, are whales also spending time and likely feeding in the same habitats that either of these two preys inhabit?

Page 18, line 8: "From an isotopic perspective, if they lived much deeper, or in a different place in the water column, we would expect their isotope ratios to reflect that different habitat." A reference is needed to support this claim in the Gulf of Alaska. Isotope depth-gradients are not often present and obvious and a recent reference demonstrating their existence in the study area is needed.

Minor edits:

Page 11, line 24: repetition of "samples"

Page 13, lines 32: Provide a brief definition of isospace

Page 23, line 45: "should be accounted" for

Decision letter (RSOS-191110.R0)

08-Oct-2019

Dear Dr Wild,

The editors assigned to your paper ("Exploring variability in the diet of depredating sperm whales in the Gulf of Alaska through stable isotope analysis") have now received comments from reviewers. We would like you to revise your paper in accordance with the referee and Associate Editor suggestions which can be found below (not including confidential reports to the Editor). Please note this decision does not guarantee eventual acceptance.

Please submit a copy of your revised paper before 31-Oct-2019. Please note that the revision deadline will expire at 00.00am on this date. If we do not hear from you within this time then it will be assumed that the paper has been withdrawn. In exceptional circumstances, extensions may be possible if agreed with the Editorial Office in advance. We do not allow multiple rounds of revision so we urge you to make every effort to fully address all of the comments at this stage. If deemed necessary by the Editors, your manuscript will be sent back to one or more of the original reviewers for assessment. If the original reviewers are not available, we may invite new reviewers.

- Data accessibility

It is a condition of publication that all supporting data are made available either as supplementary information or preferably in a suitable permanent repository. The data accessibility section should state where the article's supporting data can be accessed. This section should also include details, where possible of where to access other relevant research materials

such as statistical tools, protocols, software etc can be accessed. If the data have been deposited in an external repository this section should list the database, accession number and link to the DOI for all data from the article that have been made publicly available. Data sets that have been deposited in an external repository and have a DOI should also be appropriately cited in the manuscript and included in the reference list.

If you wish to submit your supporting data or code to Dryad (<http://datadryad.org/>), or modify your current submission to dryad, please use the following link:
<http://datadryad.org/submit?journalID=RSOS&manu=RSOS-191110>

- **Competing interests**

- **Authors' contributions**

- **Acknowledgements**

- **Funding statement**

Best regards,
Lianne Parkhouse
Royal Society Open Science
openscience@royalsociety.org

on behalf of Dr Asha de Vos (Associate Editor) and Professor Kevin Padian (Subject Editor)
openscience@royalsociety.org

Associate Editor's comments (Dr Asha de Vos):

Thank you for your submission. This paper is interesting and adds a lot of value to our understanding of whales and fisheries however based on our reviewer's comments this

manuscript requires major revisions. We hope you will find the comments useful to fine-tuning your paper. Thanks

Reviewers' Comments to Author:

Reviewer: 1

Comments to the Author(s)

The manuscript titled "Exploring variability in the diet of depredating sperm whales in the Gulf of Alaska through stable isotope analysis" uses stable isotope analysis to identify a trophic shift in sperm whales. Over the past decades they have shifted their diet to a high dependence on Sablefish, and they achieve this by depredating catches from GOA long line ground fisheries. Overall, this paper is well-written and shows understanding of the caveats associated with stable isotope analysis and mixing models. The authors have done a good job at addressing these where possible. The biggest concern is the TEFs used, as these are estimated from other species. TEFs are affected by so many variables, it is hard to be sure that the TEFs between these species are comparable. However, after a short literature review I concluded that the authors have done their best to find the most appropriate values and that those used in the manuscript are the best available options.

Nevertheless, I have identified a few points that need to be addressed before this paper is acceptable for publication.

Specific comments:

Throughout manuscript: Line numbers don't match lines

Page 1 Line 22: "157,680 sperm whales", put this into context. What is the current population, what was the estimated population then? I don't know whether to think this is a lot or not.

Page 2 Line 17: This does not need to be added to the manuscript, but I am curious, how can one tell that a whale is picking up vessel-hauling sounds from 10 miles away?

Page 3 Line 31: You mention the downsides to using fecal samples and direct observation to assess diet. Despite the low sample sizes, these studies may be the only direct insight we have to diet in recent years. You based the included prey types in your models off of stomach contents from decades ago, how do those data align with more recent studies?

Page 4 Line 8: A more recent reference: Barton et al. 2017

Page 5 Line 24: How similar?

Page 6 Line 43: Species should not be included in mixing models unless they have been observed in diets. Have sperm whales been observed feeding on these grenadier species? If not, they should be removed from the analysis.

Page 7 Line 48: What is the difference between a 1 inch chunk and a 1 inch section? This should also be in centimeters.

Page 8 Line 50: What standards were used to determine analytical precision? Were duplicate samples used to assess sample homogeneity? What reference materials were used?

Page 9 Line 8: This sentence is awkward, and grammatically incorrect. Try "Stable isotope ratios of materials change as they are incorporated into a consumer's tissues; the difference in isotope values between predator and prey is known as a Trophic Enrichment Factor (TEF)." No need to mention isotopic incorporation rates as you don't use that terminology throughout the paper.

Page 9 Line 12: Remove "primarily for $\delta^{15}\text{N}$ values"

Page 9 Line 24: Are there no estimations for carbon TEFs?

Page 10: All page numbers from here on out appear as "1" from 10-19 and "2" from 20-29, and "3" onwards.

Page 12 Line 25: What TEFs were used to calculate trophic levels of non-sperm whale species?

Page 12 Line 34: Why did you use ANCOVA for skates and ANOVA for Grenadier?

Page 13 Line 20: Isotopic ratios and Isotopic space are the same thing, instead of isotopic space I believe you mean "trophic niche space"

Page 14 Line 35: Is this supposed to say "more positive"? A more negative value suggests feeding closer to shore.

Page 16 Line 43: One possible explanation that isn't addressed is energetic requirements. Are the more frequent depredators larger? Fatter? Healthier than the less frequent depredators? Perhaps the more frequent ones are seen more frequently because they spend more time foraging than the others, and therefore when longlines are not available they continue to feed on skates.

Page 18 Line 6: remove "in"

Page 21 Line 12: The lipid rich aspect is an interesting one. If the data is available, would it be possible to compare the lipid content/energy content of the prey items. This could suggest whether these whales have shifted to a better prey resource, but more importantly speaks to ecosystem function rather than just food web structure. If sablefish are high energy fish, and this new trophic pathway is changing energy flow pathways so that it is bypassing other dependent predators, this could have significant impacts on ecosystem functions.

Figure 3: The 5 mixtures that branch off of the main cluster that have decreased N and C values, were those from any particular region? Looking at the sampling map it appears as though approximately 5 whale samples were taken in different locations than the rest, and I wondered if these were those samples. If so, I would be interested to see if predicted dietary contributions would change if these outliers were removed.

Reviewer: 2

Comments to the Author(s)

This is very interesting paper which does a good job at demonstrating the impact that disappearing prey and increasing fisheries has had on sperm whale diets and really comes together in the discussion. After some improvements, corrections and edits, I believe this paper will be of great value to our understanding of fisheries' impacts on the ecology of marine organisms and ecosystems.

I would suggest reorganising the introduction, notably by switching sections 1.2 and 1.3 around to provide a more streamlined understanding of the study subject. Start with the general theme of sperm whale diet and history and narrow down to the subject of depredation which will aid the flow leading to the goals described in 1.5.

Page 8, line 24: Two samples from whales were taken outside of the study area but included due to similar isotope values. However, Page 10, line 8, specimens of prey sampled outside of the study area were not included in the final analysis. Inconsistency of methodology?

Page 9, Line 29: Two groups of skates are chosen in the analysis: "longnose skates" and "other skates".

1. It is stated that "other" skates are most likely to be Alaskan and Big skates (Page 9, line 33), what is the evidence for this?
2. These samples come from fisheries' bycatch, were they identified by fishermen? If so, fishermen are likely to misidentify and mislabel skates which are notoriously prone to confusion (Iglesias et al. 2009; Dulvy et al. 2000). It is likely that among "other skates" were also counted longnose skates, in that case their isotopic ratios may have skewed the "other" group's isotopic ratio towards non-significance. It would be interesting and pertinent to see that variability in isotope ratios within the group does not suggest different species with different ratios and thus different contributions to the sperm whales' diets.

Page 12, line 31: "While bottlenose dolphins are much smaller than sperm whales, they are a toothed cetacean that consumes fish as prey that may have trophic levels comparable to sperm whale prey." This is a highly speculative statement, what is the evidence for their trophic levels being comparable?

Page 14 hypothesis 1: A major objective of the paper is to question whether the diet of sperm whales has changed over the past 15 years, but the results don't present an analysis of the trophic level of sperm whales between years. If possible, I would be interested in seeing a year to year comparison of the isotope data to present any gradual change over time of sperm whale diet.

Page 14, Hypothesis 2: The hypothesis questions seasonal changes in sperm whale diet, however sampling is only carried out over 1 season, how can seasonal variability be assessed? I would suggest reformulating the hypothesis to state that the variability considered here is within a season and not among seasons.

Page 14, line 34: some whale individuals are classed as "frequent" depredators. The methodology does not indicate how depredation is assessed, besides being sighted in proximity to longlines, what is the guarantee that these whales are actually depredating on lines?

Page 15, line 15: I would recommend changing the titles of subsections from "Objective x" to the actual objective description for ease of reading.

Page 15, line 42: "The $\delta^{13}\text{C}$ and $\delta^{15}\text{N}$ of magister squid and the $\delta^{13}\text{C}$ values of spiny dogfish differed significantly among depth strata (Table 3). Nevertheless, because all depth strata were within sperm whale habitat, we considered all specimens of a given species to be representative of the available prey and combined them into a single group across depth strata for use in mixing models."

All depths strata were combined into a single group because they were all within sperm whale habitat. But with if prey item isotope values were significantly different by depth strata, it would be interesting to see an analysis by depth strata to understand if whales are feeding at specific depths and how this might have change over the past 15 years with the onset of depredation.

Page 18, line 20-40: Prey items were grouped due to their similar isotopic niche. Justification was given for sablefish likely being the largest contributor of the 2 preys. However, a similar isotopic niche does not equate to a similar trophic or environmental niche. Does our knowledge of these prey items' ecology and that of sperm whales allow us to infer which prey may be contributing most? i.e. habitat use of sperm whales vs habitat use of prey. We know that sablefish are depredated on by whales, are whales also spending time and likely feeding in the same habitats that either of these two preys inhabit?

Page 18, line 8: "From an isotopic perspective, if they lived much deeper, or in a different place in the water column, we would expect their isotope ratios to reflect that different habitat." A reference is needed to support this claim in the Gulf of Alaska. Isotope depth-gradients are not often present and obvious and a recent reference demonstrating their existence in the study area is needed.

Minor edits:

Page 11, line 24: repetition of "samples"

Page 13, lines 32: Provide a brief definition of isospace

Page 23, line 45: "should be accounted" for

Author's Response to Decision Letter for (RSOS-191110.R0)

See Appendix A.

RSOS-191110.R1 (Revision)

Review form: Reviewer 1

Is the manuscript scientifically sound in its present form?

Yes

Are the interpretations and conclusions justified by the results?

Yes

Is the language acceptable?

Yes

Do you have any ethical concerns with this paper?

No

Have you any concerns about statistical analyses in this paper?

No

Recommendation?

Accept as is

Comments to the Author(s)

Thank you for addressing my comments. My main concern with this article was (and still is) the lack of recent diet information. Since mixing models depend on a priori knowledge of prey items, you had to make a lot of assumptions to do this analysis. However, it is better to include extra prey items than miss prey items, so I think your approach is warranted and is the most appropriate way to deal with this data. I have no further comments.

Decision letter (RSOS-191110.R1)

27-Jan-2020

Dear Dr Wild,

It is a pleasure to accept your manuscript entitled "Exploring variability in the diet of depredating sperm whales in the Gulf of Alaska through stable isotope analysis" in its current form for publication in Royal Society Open Science. The comments of the reviewer(s) who reviewed your manuscript are included at the foot of this letter.

At this stage, we ask that you please archive your GitHub code within the Zenodo repository: <https://guides.github.com/activities/citable-code/>. By doing this, a formal, citable DOI will be associated with your data record, and an open license (CC-BY preferred) can be applied to your data. Please can you ensure this is done, and you send the details of the Zenodo accession - we can then update your data access statement to include a sentence along the lines below:

"Relevant code for this research work are stored in GitHub: [GitHub URL here] and have been archived within the Zenodo repository: <https://doi.org/zenodo.....> [ref number]."

on behalf of Dr Asha de Vos (Associate Editor) and Kevin Padian (Subject Editor)
openscience@royalsociety.org

Associate Editor Comments to Author (Dr Asha de Vos):

Associate Editor: 1

Comments to the Author:

It seems you have worked hard to address the concerns outlined by our reviewers. Thank you and I look forward to the final product.

Reviewer comments to Author:

Reviewer: 1

Comments to the Author(s)

Thank you for addressing my comments. My main concern with this article was (and still is) the lack of recent diet information. Since mixing models depend on a priori knowledge of prey items, you had to make a lot of assumptions to do this analysis. However, it is better to include extra prey items than miss prey items, so I think your approach is warranted and is the most appropriate way to deal with this data. I have no further comments.

Appendix A

Dear Dr. Parkhouse, and Editor de Vos,

Thank you for allowing us to submit a revision of this manuscript for your review. We hope you find the manuscript revised sufficiently for publication, and please do not hesitate to let us know if there are additional changes or edits needed.

Comments from another outside reviewer prompted us to re-organize the Objectives and Hypothesis throughout the methods and results sections so that the reader could follow along more easily. Reviewer #1 alluded to the issues as well, which justified making these changes. This additional reviewer asked us to switch a few paragraphs in the discussion (though not the content) and directed us to add a few additional calculations to the results sections of Hypotheses 1 and 3. We believe the manuscript is now stronger, and hope these changes are acceptable and not too far out of the original manuscript's flow.

A detailed response to each reviewer's comments follows:

Responses to Reviewer # 1:

Throughout the manuscript line numbers don't match lines.

This happened when the manuscript was uploaded into Manuscript Central and we also thought it was odd how the line numbers were not added to match the manuscript when it was turned into a PDF but were unable to figure out a way to fix it. We will work with the editors to make sure it is corrected.

Page 1, Line 22: Putting the 157,680 sperm whales killed during Russian whaling into context.

We have added the overall estimated population size at the time (1.2million). The current population is unknown and no reliable estimate exists for the North Pacific. However, there are some area-specific estimates in the Gulf of Alaska that we have added to the manuscript.

Page 2, Line 17: "This does not need to be added to the manuscript, but I am curious, how can one tell that a whale is picking up vessel-hauling sounds from 10 miles away?"

We had a group out satellite tagging sperm whales, while another group of researchers was on-board a longline fishing vessel that had a few sets in the water. The tagging group was working around a few animals that were generally milling about, when all of the sudden the animals all beelined northward. The tagging vessel noted the location, and tracked the animals northward, until they came to the fishing vessel which had just begun its fishing haul. Coincidentally it was the vessel the other researchers were on, so locations and haul timing were recorded, as well as acoustic data from recorders in the area. There have been a few instances similar to this we have experienced while conducting research where we were able to confirm that individual whales abruptly changed course or moved quickly toward a vessel that had just begun to haul a fishing set.

Page 3, Line 31: Regarding the downside to fecal samples and direct observation to assess diet. Despite the low sample sizes these studies may be the only direct insight we have into diet in

recent years. You based the included prey types in your models off of stomach contents from decades ago; how do those data align with more recent studies?

We apologize for not being clearer. We were not able to find any fecal studies or stomach contents from strandings in the last 20 or 30 years in or around the GOA. Given the very different nature of diet around the world, we felt that we should not use more recent stranding stomach information from, say, the eastern Tropical Pacific, or North Atlantic, where there have been a few strandings in the last 20 or 30 years.

The most recent stranding of sperm whales anywhere near the Gulf of Alaska, during which stomach contents were collected, was 16 June, 1979 on the coast of Oregon, USA, when 32 stomachs of 41 total sperm whales that stranded were examined (Harvey et al. 2014, Marine Mammal Science). This study was “decades ago” as well, and thus wouldn’t help us understand more recent stomachs, as the reviewer is interested in. In addition, the stranding was a little far south outside of our study area, and there were only 8 males in the group.

We have added a note in this line in our manuscript to include the “bias towards hard parts from species that are less likely to break down in the stomach and thus can be identified in stomachs or feces” as well. We have also added a sentence to reflect the fact that no stranded stomachs or fecal samples have been collected or analyzed from this region, and so, unfortunately, our most reliable data from which to collect prey for the current study remains in whaling data from decades ago.

Page 4, Line 8: Barton et al. 2017 reference added.

Page 5, Line 24: How similar?

This is with reference to the two isotope samples that were taken outside of the study area. We have revised this section in a few ways. First, this was a typo and there were three, not two samples taken outside of the study area – it has been corrected. Second, we have added in the results of a multivariate analysis of covariance (mancova) that was used to analyze the differences between regions of both isotope ratios together ($\delta^{13}\text{C}$ and $\delta^{15}\text{N}$), which revealed no significant differences between regions. Sentence now reads:

“However, these samples fell within $\delta^{13}\text{C}$ and $\delta^{15}\text{N}$ values of other samples and isotope ratios from central GOA were not significantly different than other samples (MANCOVA, $F_{8,56}=2.19$, $p=0.05$), and so they were included in the analysis to help improve our sample size.”

Page 6, Line 43: Species should not be included in mixing models unless they have been observed in diets. [...] If sperm whales are not observed feeding on grenadier, they should be removed from analysis.

We have indeed seen behavior from whales to suggest they will go after grenadier; for example, we have observed whales at the surface following closely behind a longline vessel dip underwater, while turning, after grenadier have fallen from hooks at the surface during a longline haul. We have added this to the manuscript. In addition, as we stated in the manuscript, grenadier are readily available to sperm whales on hooks coming up with longline fishing gear. We recognize the dilemma that species should not be included in mixing models if they aren’t part of the diet. However, feel that due to difficulty observing diet, coupled with

lack of stomach contents in recent history from this region, we cannot rule out a species that is abundantly available as bycatch on fishing gear that is targeted during depredation activities. It makes us uneasy to remove grenadier from the analysis, as it feels like we would then be, in a way, cherry picking the data from which our mixing models have to choose sperm whale diet from. Without any information to the contrary, we must assume that any fish coming up regularly as bycatch on longline fishing gear in this region, may be taken by whales, at least in some capacity. We then leave it to the mixing models to tell us whether this species is indeed a substantial contributor to diet of sperm whales (in this case, it is not!).

Page 7, Line 48: Inches have been changed to cm, and “chunk” vs. “section” resolved.

Page 8, Line 50: What standards were used to determine analytical precision? Were duplicates used to assess sample homogeneity? What reference materials were used?

Reference materials were indeed listed on Page 8, Line 48: Vienna PeeDee Belemnite and atmospheric N₂. These were the standards used to determine analytical precision, and we have added the sample size for duplicates used in this determination as well.

Page 9, Line 8: Suggestion for rephrasing an awkward sentence;

Thank you for the suggestion; we have re-worded the sentence accordingly.

Page 9, Line 12: Remove “primarily for $\delta^{15}\text{N}$ values”.

Done

Page 9, Line 24: Are there no estimates for carbon TEFs?

Good point. Yes there are estimates for carbon TEFs which we had left out and have now added in. As shown later, they were all used in our calculation of carbon TEF for this study (0.96‰), but as the reviewer points out, in this particular sentence we left out mention of them and only listed $\Delta \delta^{15}\text{N}$ values. We have revised the sentence to add the $\Delta \delta^{13}\text{C}$ values too.

Page 10: All page numbers from here on out appear as “1” from 10-19 and “2” from 20-23, and “3” onwards.

Interesting, we did not notice this, but you are correct. Page numbers look fine in the document prior to uploading to Manuscript Central, so it must have happened at that point in the submission process and was out of our control. We will work with the editors on the revision submission to make sure Page Numbers (and line numbers for that matter) are better listed and actually line up. We have no idea how it happened.

Page 12, Line 25: What TEFs were used to calculate trophic levels of non-sperm whale species?

A sentence has been added in the methods under the *Trophic Level Calculations* section to reflect that TEFs commonly used in the literature for marine species (including fish) were used for TL calculations of fish and squid ($\Delta \delta^{15}\text{N} = 3.5\text{‰}$).

Page 12, Line 34: Why use ANCOVA for skates and ANOVA for grenadier?

This was a typo, an ANCOVA was used for each one, and we have fixed this in the text!

Page 13, Line 20: Isotope ratios & isotopic space are same thing; instead of isotopic space I believe you mean “trophic niche space”

This is true, thank you for catching that. We have edited the sentence to use “trophic niche space” rather than isotopic space.

Page 14, Line 35: Is this supposed to say “more positive”? A more negative value suggests feeding closer to shore.

We believe we got this one right originally. Higher $\delta^{13}\text{C}$ values, which are LESS negative, reflect inshore feeding, while LOWER $\delta^{13}\text{C}$ values that are MORE negative, reflect offshore foraging. See Hobson et al. 1994 (Figure 1), Sydeman et al. 1997 (Figure 7).

Page 16, Line 43: One possible explanation that isn't addressed is energetic requirements; are the frequent depredators larger? Fatter? Healthier? Than less frequent depredators.

Interesting idea - unfortunately we have no measurements of body condition or size of these animals, except what we notice while we're on the water. Two of our frequent depredators actually seem to be very different sizes – they have been seen together a lot over the years, and one looks noticeably bigger than the other (though they are both adult males).

We did think about the caloric value of skates versus other fish and squid, which is similar to what the reviewer is getting at here [see comment below]. There isn't much calorimetry data publicly available on adult skates, sablefish, rockfish, dogfish, etc. but we have a colleague at NMFS who had some bomb calorimetry data of adult sablefish and [a very small sample size of] some miscellaneous skates. The data essentially showed sablefish have a much higher percent lipid than skates, but doing a full review of lipids and calorimetry of each species was outside the scope of this study. We have added a note to the discussion to acknowledge that further work to look at this in more detail could illuminate more information about prey preferences in sperm whales.

Page 18, Line 6: Remove “in”.

Done

Page 21, Line 12: Regarding the lipid-rich aspect of depredation of sablefish. Would it be possible to compare the lipid content/energy content of prey items? This could suggest whether these whales have shifted to a better prey resource, but more importantly speaks to ecosystem function rather than just food web structure. If sablefish are high energy fish, and this new trophic pathway is changing energy flow pathways so that it is bypassing other dependent predators, this could have significant impacts on ecosystem function.

See response from Page 16, Line 43, above: unfortunately that data was not available to us and is outside the scope of this study. NMFS does have lipid content data on sablefish, but not much consistently on the other adult prey species in this study for a full comparison. In chatting with some of our colleagues at NMFS, we have been able to gather that some calorimetry analysis was done on skates, though it seems to include only a few Alaskan skates and no Longnose or Big skates. While extremely interesting, we feel it is outside what we could do for this study but is definitely a lead that should be examined in the future! We have noted this in this discussion

section as an avenue for future analysis to elucidate how new trophic pathways may impact ecosystem function and be changing energy flow pathways.

Figure 3: The 5 mixtures that branch off [in the lower left] – were they from any particular region? Looking at the sampling map it appears as though approximately 5 whale samples were taken in different locations than the rest, and I wonder if these were those samples. If so, I would be interested to see if predicted dietary contributions would change if these outliers were removed.

Thank you for noticing these 5 samples. We did too, and we looked at year, month, region, etc. to see if they had any particular characteristics. They were not taken from a different region (they were offshore Baranof & Chichagof Islands, along with a majority of the other samples). They were collected in June and July (the most popular months for samples) of 2004, 2016, 2017. Four were non-frequent depredators and 1 was an unknown ID so we don't know if it was a frequent or non-frequent depredator. In the end, we did look into these samples, but couldn't find anything unique about them that would give us any justification for removing them as outliers or treating them differently. Our guess is that potentially those whales had recently arrived in Alaska or preferred to eat something that made them occupy a different trophic space than the others, but something we couldn't tease out given the information we had available.

We did add a sentence in the results section of the manuscript to address that we looked into these samples: "Five mixtures branched off the main cluster with decreased $\delta^{13}\text{C}$ and $\delta^{15}\text{N}$ values (Figure 3), however they did not come from a specific year, month, or region that would separate them out from the other mixtures in this analysis."

Thank you to Reviewer #1.

Responses to Reviewer # 2:

Reviewer # 2 suggests reorganizing the introduction and switching sections 1.2 and 1.3 to provide a more streamlined understanding of the study subject. We have done this, which hopefully helps the flow of the introduction.

A note on page numbers for Reviewer # 2; they are consistently different from where we found the actual phrases. We're not sure where the miss-communication of page numbers happened, but it happens throughout Reviewer # 2's comments, making it difficult at times to find the appropriate sections. Line numbers did match up though, allowing us to find referenced lines eventually in most cases.

Page 8, Line 24: Two samples from whales were taken outside of study area, but included due to similar isotope values; However, Page 10, Line 8 states specimens of prey sampled outside the study area where not included in the final analysis. Inconsistency of methodology?

We think this is actually Page 5, Line 24? And “Page 10, Line 8” refers to “Page 7, Line 8”. Thank you for bringing this to our attention. We neglected to mention that the three prey samples outside of the study area were also in very shallow, nearshore waters, which we neglected to mention in the manuscript. They were also the only samples of that prey species, which sets them apart from the whale samples outside of the study area. Combined with such a low sample size (n=3), we ultimately chose not to use them in the analysis. We have adjusted the text to reflect this, adding that they were in shallow waters, not sperm whale habitat, and had a very low sample size. Finally, as we note, we did run the analysis with these 3 specimens included and they did not alter the results so we felt confident in leaving them out for simplicity.

We have also added a multivariate analysis of the whale samples to show they were not significantly different than the other whale samples and that they improve our sample size for whales.

Page 9, Line 29: Two groups of skates are chosen in the analysis: “longnose skates” and “other skates”.

We assume the reviewer is referring to Page 6, Line 29.

- 1) It is stated that “other” skates is probably Alaskan & Big skates; what is the evidence for this?

The skates were collected by NMFS personnel on the survey, and these NOAA scientists told us that while they didn’t record any skates to species other than longnose, they primarily caught Alaskan and Big skates when they weren’t catching longnose skates. We have added text in this section to reflect that.

- 2) These samples come from fisheries bycatch, were they identified by fishermen? If so, they are probably wrong.

We only collected the squid species and a few sablefish samples from commercial fishermen, so we went back to the manuscript to clarify this and realized we had mislabeled the sources in Table 1. Thanks for inadvertently catching that! All skates, grenadier, rockfish, and dogfish, along with most sablefish, came from the NMFS Longline Survey. This has been corrected in Table 1 and better referenced within the text as well.

Page 12, Line 31: “While bottlenose dolphins are much smaller than sperm whales, they are a toothed cetacean that consumes fish as prey that may have trophic levels comparable to sperm whale prey.” Highly speculative, what is the evidence for this?

Actually Page 9, Line 31. Point taken here. Bottlenose dolphin prey are likely a step lower in the food web (i.e. herring, capelin, sprat). We had been trying to get at the idea that though bottlenose dolphins are smaller than sperm whales, they feed higher trophically than other more similar-sized cetacean species, such as fin whales, and thus there can be an argument for comparing their TEFs to that of sperm whales. We have removed this sentence from the manuscript to avoid speculation, though.

Page 14, Hypothesis 1: A major objective of paper is to question whether diet of whales have changed over past 15 years, but no analysis of TL of sperm whales between years. I would be interested to see a year-to-year comparison of the isotope data to present any gradual change over time of sperm whale diet.

Actually Page 9/10, Hypothesis 1. We have added a brief analysis to the results using multivariate analyses to address variability in isotope ratios among years. We do in fact find significant differences in Year in the multivariate analysis, which when isolated to univariate analyses show the $\delta^{13}\text{C}$ values to be significantly different among years ($p=0.02$) compared to $\delta^{15}\text{N}$ values ($p=0.08$). When you look at annual variability over time (plots below), the $\delta^{13}\text{C}$ values fluctuate each year, but there is no trend over time. We must also acknowledge the small sample size per year (shown over boxes in the left-hand plot), and the potential for individual variability to influence those results. Larger sample sizes would help tease this out in the future.

Page 14, Hypothesis 2: The hypothesis questions seasonal changes in sperm whale diet, however sampling is only carried out over 1 season; how can seasonal variability be assessed? I would suggest reformulating hypothesis to state that variability considered here is within season and not among season.

Done. Actually Page 11

Page 14, Line 34: Some whale individuals are classified as “frequent” depredators. The methodology does not indicate how depredation is assessed, besides being sighted in proximity to longlines; what is the guarantee that these whales are actually depredating on lines?

Actually Page 11, Line 34. Proximity to fishing vessels is the most conclusive and worldwide adopted metric for assessing whether or not sperm whales are depredating. Interestingly, the 500m metric for proximity for depredation was achieved independently and nearly simultaneously in both the Gulf of Alaska (Mathias et al. 2012) and in the Crozet Islands (Tixier et al. 2010, Roche et al. 2007), and in South Georgia (Ashford et al. 1996). We have added a sentence to our introduction to further define depredation as such. Thus while there is always a chance that whales surfacing within 500m of a vessel are simply foraging naturally around the fishing gear, given our combined experience with these interactions and our close collaboration with the fishing industry, that whales repeatedly surfacing within 500m of a fishing vessel while it is hauling longline gear are most likely to be engaging in depredation behavior.

Page 15, Line 15: I would recommend changing the titles of subsections from “Objective X” to the actual objective description for ease of reading.

Done. We have also added Objective references in the methods and results to highlight them better in the flow of the paper.

Page 15, Line 42: The $\delta^{13}\text{C}$ and $\delta^{15}\text{N}$ of magister squid and $\delta^{13}\text{C}$ of spiny dogfish differed significantly among depth strata (Tbl 3). Nevertheless, because all depth strata were within sperm whale habitat, we considered all specimens of a given species to be representative of the available prey and combined them into a single group across depth strata for use in mixing models. If prey item isotope values were significantly different by depth strata, it would be interesting to see an analysis by depth strata to understand if whales are feeding at specific depths and how this might have changed over the past 15 years with onset of depredation.

Actually Page 12, Line 42: In addition to our reasoning that all depth strata were within sperm whale habitat and foraging availability, we kept all specimens of a single species together because of their similar isotopic values. We have added some text to reflect this. Unfortunately, we do not have depth estimates for some of the *Onykia* squid that were donated by fishermen, and some of the sablefish donated by fishermen and so we could not do a full analysis of isotope ratios by depth for the prey, as we would have liked, in a multivariate analysis. Mixing models care more-so about how similar each “group’s” isotope ratios are, than if the animals that make up the group are similar in other ways (trophic niche, environmental niche, etc.) so we are limited in how we enter prey items to the scope of mixing models themselves. Thus we were not able to use mixing models to determine whether or not different whales fed on different size or depth gradients of prey, unfortunately. Given the similarities in isotope ratios of the magister squid amongst different depths, and the dogfish among depth strata, we ultimately chose to group them together. This was strengthened from a biological perspective in that they are generally available to sperm whales in the same habitat. Unfortunately, their isotope ratios are too similar and overlapping at the different depth strata for mixing models to tease out differences in sperm whale foraging within the species themselves. Additionally, as we stated, it was necessary to reduce the prey groupings into fewer end members for the model to reduce uncertainty. We did explore separating out into a few depth strata, as the reviewer suggests, but the uncertainty in estimates was too high to really say much with regard to which depth strata the whales were feeding at, unfortunately. We agree it would be very interesting if we could tease out depths whales are targeting and whether or not that has changed, but given the similarities in so many of the prey items in general, and the fact that magister squid doesn’t really seem to make up a high proportion of their diet either way, we were unable to do that in this analysis. We have added some text to reflect the choice to keep each prey together due to isotopic similarities (though significantly different by depth strata, still very similar to each other compared to other prey end members), in addition to availability to sperm whales, and that grouping prey to reduce end members was necessary to reduce uncertainty in the model as well. We hope this explanation suffices for the reviewer.

Page 18, Line 20-40. Prey items were grouped due to their similar isotopic niche [sablefish & dogfish]. Justification given for sablefish likely being the largest contributor of the 2 preys.

However, similar isotopic niche doesn't equate to similar trophic or environmental niche. Does our knowledge of these prey items' ecology and that of sperm whales allow us to infer which prey may be contributing most? i.e. habitat use of sperm whales vs habitat use of prey. We know that sablefish are depredated on by whales; are whales also spending time and likely feeding in the same habitats that either of these two preys inhabit?

[Actually Page 15, Line 20-40]. The reviewer is correct that we cannot equate isotopic niche to trophic niche or environmental niche, and we have tried to be careful not to do so. With respect to environmental niche, we have added a note into this section of the discussion to address what is known about trophic ecology of these two prey species, notably that they inhabit very similar habitats, and are naturally going to be readily available to whales. From fishermen conversations we know that dogfish tend to "school", though there is little published research on adult dogfish in the North Pacific. We do note in the manuscript that based on biomass, sablefish would be more prevalent and thus potentially a more viable food source than dogfish. As for what we know about each prey's diet, stomach contents data shows they both consume a large amount of fishery offal, but that there are some differences in diet. As the reviewer notes, the two species may not inhabit the same "trophic niche". However, we are not arguing that they do, in this section. It is outside the scope of this work. We *can* acknowledge that they occupy similar isotopic space, and beyond that we must conjecture whether or not we can figure out which species sperm whales may be foraging more on, if any. We believe we make a strong case for sablefish, given what we know about their overlapping habitat, differences in biomass, and depredation preference of sablefish. Given that sablefish and dogfish have such similar isotopic signatures, inhabit overlapping habitat, are similar sized, and have some similarities in stomach contents data, they likely also do have similar trophic niches.

Page 18, Line 8: "From an isotopic perspective if they lived much deeper, or in a different place in the water column, we would expect isotope ratios to reflect that different habitat." A reference is needed to support this claim in the GOA. Isotope depth-gradients are not often present and obvious and a recent reference demonstrating their existence in the study area is needed.

This is a valid point, and that statement now has references from the GOA and Bering Sea region, Alaskan rockfish, and GOA primary consumers.

Minor edits:

Page 11, Line 24: Repetition of "samples"

We found this line to read: "Hypothesis 3: Sperm whale diet composition was the same between frequent and non-frequent depredators". We did a word search and did not find the word "samples" to be repeated anywhere. Given the other comments from this reviewer, this should be found on page 8, line 24, so we removed one of the "samples" from that sentence.

Hypothesis 2 (Page 11, Line 15-19) also has the word 'sample' in it a lot, so we paired that down.

Page 13, Lines 33: Provide a brief definition of isospace;

Done.

Page 23, Line 45: "should be accounted" for

Done; though we found this to be on Page 20, Line 45, not Page 23?

Thank you to Reviewer #2

Sincerely, Wild et al. authors